# Effects of Hydrostatic-Pressure on Muscle Contraction: A Look Back on Some Experimental Findings

**DOI:** 10.3390/ijms24055031

**Published:** 2023-03-06

**Authors:** K. W. Ranatunga, M. A. Geeves

**Affiliations:** 1School of Physiology, Pharmacology & Neuroscience, University of Bristol, Bristol BS8 1TD, UK; 2Department of Biosciences, University of Kent, Kent, Canterbury CT2 7NJ, UK

**Keywords:** muscle force, crossbridge force, pressure sensitivity, actin–myosin crossbridge cycle

## Abstract

Findings from experiments that used hydrostatic pressure changes to analyse the process of skeletal muscle contraction are re-examined. The force in resting muscle is insensitive to an increase in hydrostatic pressure from 0.1 MPa (atmospheric) to 10 MPa, as also found for force in rubber-like elastic filaments. The force in rigour muscle rises with increased pressure, as shown experimentally for normal elastic fibres (e.g., glass, collagen, keratin, etc.). In submaximal active contractions, high pressure leads to tension potentiation. The force in maximally activated muscle decreases with increased pressure: the extent of this force decrease in maximal active muscle is sensitive to the concentration of products of ATP hydrolysis (Pi—inorganic phosphate and ADP—adenosine diphosphate) in the medium. When the increased hydrostatic pressure is rapidly decreased, the force recovered to the atmospheric level in all cases. Thus, the resting muscle force remained the same: the force in the rigour muscle decreased in one phase and that in active muscle increased in two phases. The rate of rise of active force on rapid pressure release increased with the concentration of Pi in the medium, indicating that it is coupled to the Pi release step in the ATPase-driven crossbridge cycle in muscle. Pressure experiments on intact muscle illustrate possible underlying mechanisms of tension potentiation and causes of muscle fatigue.

## 1. Introduction

An active muscle contraction is thought to involve a relative sliding between the thin (A = actin) filaments and the thick (M = myosin) filaments in a sarcomere; this basic idea was proposed ~70 years ago and is referred to as the sliding filament theory (see review by Ranatunga [1]). The underlying, driving process is a repetitive interaction of myosin heads (crossbridges) in thick filaments with thin (actin) filaments; a crossbridge attaches to actin, undergoes a conformational change generating muscle force (and power), and then detaches. This mechanics cycle is coupled to an enzymic reaction, hydrolysis of ATP by actomyosin ATPase: thus, energy liberated during the release of the products of ATP hydrolysis (phosphate = Pi, and adenosine diphosphate = ADP) gets converted into work (and heat). Thus, an active muscle is “a machine converting chemical energy to mechanical energy”. Despite many investigations, and using different techniques, exactly how various steps in these two cyclic processes, chemical and mechanical, are coupled during an active muscle contraction, remains not fully understood. Moreover, in an intact muscle, the contractile process is more complex, it is initiated by cell membrane electric-potential changes, Ca release, etc., and how the system operates with respect to changes in temperature, and pressure remains exciting but incompletely understood (see Geeves and Ranatunga [2]: Ranatunga [1] and refs). 

It has been known for nearly 100 years (see Cattel and Edwards, [3,4], Brown [5,6]) that skeletal muscle contractions are sensitive to changes in hydrostatic pressure in the surrounding medium. 

(i).Looking back on some experimental studies made on intact vertebrate muscles, the findings show that increased pressure produces marked twitch tension potentiation (see Vawda et al. [7] and references therein). With the twitch contraction being dependent on excitation, Ca release, and uptake, etc., sorting out the underlying events that lead to tension potentiation can be complex. Interestingly, such effects were slight in tetanic contractions. Some other basic findings in relation to the isometric muscle (fibre) are the following (see Ranatunga et al. [8]).(ii).In a resting/relaxed fibre, the tension is insensitive to pressure (rubber-like behaviour). (iii).In a rigour muscle, the tension increases with pressure (as in a normal elastic element like glass). (iv).In a maximally activated muscle fibre, the tension gets depressed by high pressure and the extent of tension depression is sensitive to presence of the products of ATP hydrolysis, Pi, and ADP (Fortune et al. [9]).(v).In biochemical experiments—in solution, it has been shown that the interaction between isolated actin and myosin subfragment, is pressure-sensitive (Coates et al. [10]). Consequently, the effects of increased hydrostatic pressure on mechanics of muscle and fibres became interesting to examine results in relation to biochemical findings. (vi).Use of rapid pressure release as a perturbation technique also has become a useful technique to examine the underlying molecular mechanisms of force generation (see [2,11]), fatigue [12] and excitation–contraction coupling in muscle [13].

The main aims of this short review are the following:(a)To re-examine the basic findings in such pressure perturbation studies made on muscle fibres;(b)To emphasize aspects that are not fully understood;(c)To highlight some ideas for future thinking.

A reader interested in more specific details should read the original papers listed here. 

## 2. Materials, Methods, and Techniques

The pressure chamber used in most of the experiments is described fully in the paper by Ranatunga et al. [14]. The chamber had a volume of 3 mL (1 × 1 × 3 cm) milled in a stainless steel block of 5 × 7 × 14 cm (see Figure 1A). Along its horizontal plane were seven ports; two of them were inlet/outlet ports for solution exchange, and a third was connected to a high-pressure line from a High-Performance Liquid Chromatography pump (302 pump with 802C control unit; Gilson Medical Electronics, Villiers-le-Bel, France), and a fourth was used to house the pressure transducer (601A; Kistler Instrument Corp., Amherst, NY, USA). The remaining three ports were for specially designed “transducer plugs”: one carried a force transducer (AE 801), the second one carried a micrometre for length adjustment, and the third was for temperature control/monitoring. The force transducer element was glued to a stainless steel tube that could be pressure-sealed; the beam of the AE 801 element was enclosed in a close-fitting glass tube filled with paraffin for electrical insulation (see [2]). The top and the bottom faces of the pressure chamber were fitted with Perspex (ICI Plastics), windows for illumination and visualization of the fibre, and the chamber could be opened for fibre mounting by removing the top window (see Figure 1B). In experiments investigating the effects of rapid pressure release, an electromagnetically triggered, spring-loaded valve was placed on the high-pressure input line close to the fibre chamber. The design of this valve was based on that of Davis and Gutfreund [14]. 

## 3. Results

Based on the characteristics of their force, three functional and mechanical states may be recognized in skeletal muscle; they are (i) the resting (or relaxed) state; (ii) the rigour state; and (iii) the active state. Interestingly, these three mechanical states also differ with respect to the pressure effect on their tension [8], and also, the temperature effect on their tension (see [15]). 

(i).*Resting state*: In resting state, the cross-bridges remain detached, and force develops on stretching beyond its rest length; this resistance to stretch arises from the stretch of non-crossbridge structures in sarcomeres, such as titin filaments that connect the thick filament ends to the Z-discs in sarcomeres. The resting muscle tension is largely insensitive to pressure and temperature changes. Hence, the resting muscle force is considered to be “rubber-like”. Upon stretching, a resting muscle can develop a type of “active” force, due to heat contracture at high temperatures (see Ranatunga, [15]). (ii).*Rigour state*: In rigour, with no ATP, the cross-bridges remain attached to actin, but they are not cycling. Muscles are stiff, and the force decreases slightly with increase in temperature (exothermic) or pressure; this is like in a normal elastic element, such as glass. (iii).*Active state*: In active muscle, crossbridges are cycling, i.e., they attach to actin, develop force, and detach, and, as presented below, active force rises with temperature (the process is endothermic) but decreases with increase in pressure. It is interesting to re-examine the basis of such experimental findings: the pressure effects in relation to some are listed below.

### 3.1. Pressure Effects on Different Fibre Types 

Much knowledge is available on the different mechanical behaviour of various solid materials under high pressure (see Brandes, [16]). Some experimentally determined effects of increased hydrostatic pressure on the isometric tension of elastic fibres of known mechanical characteristics (e.g., glass, rubber) as well as of other biological materials (e.g., keratin, silk, collagen) are presented here for comparison with muscle fibre responses. 

Some experimental records we previously published (see Ranatunga et al. [8]) are shown in Figure 2, Figure 3, Figure 4 and Figure 5. Figure 2 shows the effect of increased hydrostatic pressure on the tension developed by glass, collagen, and rubber (each fibre was stretched initially to maintain a steady “resting” tension). It is seen that in glass and collagen, the tension increased in phase with pressure and the tension returned to the initial value on pressure release to atmospheric. In contrast, the tension in rubber changed little with increased pressure. These records illustrate two useful points. Firstly, different tension changes are shown by different fibre types under similar experimental conditions; this indicates that the tension changes observed reflect properties of the fibres themselves and not just in the series-elasticity (e.g., glue and hooks used for attachment). Secondly, the pressure sensitivity of force decreased in sequence from glass > copper > rubber—as expected, on a theoretical basis (see Table 2 and discussion in [8]). 

### 3.2. Force in Skinned-Muscle Fibres

Figure 3 shows experimental tension records from a bundle of skinned muscle fibres on exposure to higher pressures; they show qualitatively different tension behaviours depending on the muscle fibre state. The tension of resting muscle is insensitive to pressure, rubber-like (A); force in fibre bundle in rigour rises with pressure (B), as in normal elastic matter; and force in maximally activated muscle decreases on exposure to higher pressure (C). Additionally, in all cases, the force changes by an increase in pressure are reversible on the release of pressure.

Figure 4 shows that rigour tension increases linearly with an increase in pressure, also when examined at different initial rigour tensions, the pressure/tension plots are approximately parallel, having a similar slope. In behaviours of rigour fibre (with crossbridges attached and in rigour state—not cycling), the tension is much like in a normal elastic fibre.

### 3.3. The Crossbridge Cycle: Pi, ADP Effects

The interaction between actin and myosin in solution has been shown to be pressure-sensitive (Geeves and Gutfreund, [17]) and pressure perturbation has been used to investigate the kinetics of actomyosin interaction [10]. It was therefore of interest to study the effects of hydrostatic pressure on tension development in muscle fibres. In confirmation of early work on intact whole muscles [5], our experiments showed that exposure to high pressures (up to about 20 MPa or 200 atm) produced no permanent deleterious effect in glycerinated muscle fibres. Additionally, the maximal Ca-activated tension was reversibly depressed on exposure to high pressure (Geeves and Ranatunga, [2]). Our experimental findings show that the pressure-induced tension depression in maximally Ca-activated glycerinated muscle fibres is sensitive to the concentrations of inorganic phosphate (Pi) and ADP present in the bathing solution. Experimental records from one fibre are shown in Figure 5; with an increase in pressure, relaxed fibre is insensitive (a, i), active fibre tension decreases (a, ii) and extent of depression is higher with Pi (a, iii). These findings support the suggestion (see Geeves and Ranatunga, [2]) that the depression of active tension by high pressure in a muscle fibre is at least partly due to an effect of pressure on crossbridges.

A direct comparison of the pressure-induced tension depression in the presence of 20 mM phosphate and in the presence of 11 mM Mg^2+^-ADP is shown in Figure 6, where the tension depression induced by 10 MPa pressure in the control solution is plotted on the abscissa against the corresponding value in either phosphate (■) or Mg^2+^-ADP (○) on the ordinate. Note that all the tension data in the presence of phosphate (■) are above the line of identity (full black line) and all those in the presence of Mg^2+^-ADP (○) are below it. Thus, the pressure-induced tension depression in maximally Ca-activated glycerinated muscle fibres is sensitive to the concentrations of inorganic phosphate and ADP present in the bathing solution: pressure effect is enhanced Pi but decreased with ADP.

### 3.4. Contractions of Intact Muscle Fibres

An intact muscle shows different types of contraction, twitch, tetanus, fused, etc., depending on the pattern and frequency of activation: hence, understanding pressure effects is not easy and challenging. However, they do provide an important insight into the underlying mechanisms and complexities involved in pressure effects on muscle. 

Figure 7 demonstrates pressure effects on twitch and tetanic contractions; it is seen that high pressure increases twitch amplitude (peak tension) whereas high pressure decreases tetanic tension.

A twitch contraction is the transient tension response to a single stimulus (+single action potential leading to the release of Calcium) and tension rises to a peak and relaxes. Figure 8A–C shows that twitch tension is potentiated on exposure to high pressure, accompanied by an increase in time to peak and half-relaxation time (see Figure 8D).

Figure 9 shows that the pressure effects on contractile tension in skeletal muscle vary with the stimulation frequency. At low frequencies when contractions are not summated, the tension is potentiated, whereas at higher frequencies that lead to contraction fusion (tetanic contractions), the tension is depressed by high pressure (see Figure 9B).

Figure 10A,B shows some fused tetanic contractions at atmospheric and at high pressure (pointed to by an arrow). It is seen that even in a fused tetanic contraction the rising phase is faster and the relaxation phase is slower at high pressure. The steady tension at plateau, however, is depressed at high pressure: the graphs of measurements are shown in Figure 10C,D. These data from intact fibres (bundles) clearly indicate that increased pressure has effects on several processes within muscle fibres, activation, plateau, and relaxation. Experimenting with these effects further would enable us to get some understanding of such processes.

### 3.5. Pressure Release on Single Muscle Fibre Responses

Experiments on single intact muscle fibres showed that twitch contractions are longer lasting at higher pressure, as in Figure 7 (See Figure 11a) and, analyses showed that the increase in peak twitch tension is linearly related to pressure (see Figure 11b). The exact nature of these effects and the underlying processes could be examined by recording the effects of rapid pressure release on tension responses, as shown by the experimental records illustrated in Figure 12.

As seen in Figure 12b, pressure release during enhanced twitch contraction does not lead to a decrease in the increased force, except when pressure is released very early and close to stimulus (Figure 12a). This shows that the effect of increased pressure acts on early processes (e.g., Ca release from sarcoplasmic reticulum) of twitch contraction; this would account for increased rate of tension rise, increased tension and slowed relaxation. These effects are basically similar to the tension potentiating effects produced by caffeine (see Vawda et al. [7]).

### 3.6. Some Observations of Pressure Effects on Muscle Fatigue 

Experiments on single intact muscle fibres could be used to examine the process of fatigue and Figure 13 shows experimental traces from one fibre, which also illustrate the effects of increased pressure on fatigue. Figure 13a shows that high pressure (5 MPa) produces just a small depression (and slowed relaxation) in normal unfatigued fibre. When tetanic contractions are elicited at short intervals and muscle gets fatigued (tension drops), increased pressure leads to tension potentiation, as seen in Figure 13c. 

Figure 14 shows plots of a full set of experimental data from one fibre. It is seen from Figure 14a,b that tension change due to high pressure is a depression in control (unfatigued) fibre (at 1/30 s stimulation frequency) but as fatigue develops and tension decreased to phase 3, it changes so that tension gets potentiated by high pressure (Figure 14b).

### 3.7. Rapid Pressure-Release Experiments 

Experiments on tetanically activated single intact muscle fibres could be used to examine the process of force generation in crossbridges. This is done through exposure to high pressure and then inducing the rapid release of pressure (<1 ms) during tension plateau and examining the force recovery. Figure 15a shows that the resting tension is largely unaffected by pressure change: the spike in the tension is probably an effect on the tension transducer output. Figure 15b is from an active fibre, where the pressure is released on the tetanic tension plateau. In addition to the spike is a tension drop (Phase 1), followed by a rapid recovery and rise (Phase 2a and 2b) to a level higher than that before the release.

Figure 16 shows a tension record from a fibre, illustrating the type of analyses that can be carried out on the tension transients when pressure is released on the tetanic tension plateau. Since tension at the plateau is less at high pressure, the tension finally rises to a higher level on pressure release to the atmospheric level.

As shown in Figure 16, the rising phase of tension transient to pressure release could be fitted with a double exponential providing rate and amplitude of phase 2a and of phase 2b. Pooled data from such analyses on several muscle fibres are shown in Figure 17 and Figure 18. Data in Figure 17a show that the tetanic tension is decreased linearly with increased pressure, both in steady-state measurements and in pressure-release experiments. Figure 17b shows that the amplitude of the initial tension drop (P1), is increased linearly with an increase in pressure.

The data analysis in Figure 18a, shows that the rates of tension rise (phases 2a and 2b) after pressure release remain unchanged with the size of the pressure-release step: their amplitudes, however, increased linearly, as shown in Figure 18b. Additionally, it is interesting to note that the rate of Phase 2a is within the range obtained for crossbridge force generation in ~1 ms length release experiments (see Huxley and Simmons, [21]): also, it does not contribute to tension rise above pre-release level (dotted curve). It is conceivable that pressure release induces a step length change in the muscle fibre, possibly due to shortening in the “tendon”: an increase in pressure stretches the tendon in relaxed muscle fibre and on pressure release “tendon shortens”. 

Figure 19 compares tension response to pressure release in rabbit-skinned fibre (Figure 19a) with that of frog fibre (Figure 19b). A prominent Phase 1 and a rapid tension recovery (Phase 2a) are not observed in the tension transient from the rabbit-skinned fibre. Phase 3 is not recorded in intact frog fibre (from Vawda et al. [20]). The short dashed curve in Figure 19b is Phase 2 in the rabbit fibre. As discussed in Vawda et al. [20], a prominent Phase 1 and a rapid tension recovery (Phase 2a) were not observed in the tension transient from rabbit skinned fibre and Phase 3 was not recorded in intact frog fibre perhaps due to the short duration of contraction.

## 4. Discussion

It has been known for nearly 100 years (see Refs in Geeves and Ranatunga, [2]) that skeletal muscle contractions are sensitive to changes in pressure. Findings from various experimental studies that examined pressure effects on intact and skinned mammalian and frog muscle fibres were referred to in this brief review; they provide some useful information relating to the processes and the underlying mechanisms of muscle contraction. 

### 4.1. The Experimental Findings

Some interesting basic findings on isometric muscle (fibre) are the following (see Ranatunga et al. [8]).

(i).In a resting/relaxed fibre, the tension is insensitive to pressure (as in rubber).(ii).In a rigour muscle, tension increased with pressure (as in a normal elastic element, like glass).(iii).In a maximally Ca^2+^-activated muscle fibre the tension gets depressed by high pressure and the extent of this active tension depression is sensitive concentration of the products of ATP hydrolysis, Pi, and ADP (Fortune et al. [9]).(iv).It is clear from experiments on intact fibres shown here (Figure 11, Figure 12, Figure 13 and Figure 14) that pressure effects can be complex, but in some cases, e.g., during fatigue and twitch potentiation, they provide useful insight into the underlying mechanisms. Thus, muscle fatigue can be due to the accumulation of phosphate and interfering with Ca release. Twitch potentiation on exposure to high pressure is due to enhanced Ca release on excitation.(v).Tension responses to rapid pressure release are also (Figure 14, Figure 15, Figure 16, Figure 17, Figure 18 and Figure 19) rather complex but they also provide some interesting clues regarding the underlying events.(vi).Although definitive answers are not clear, possible general mechanisms are worthy of consideration. A fast pressure release essentially produces a simultaneous ‘length step’ within muscle fibre due to the decompression (expansion) of series elasticity. This is best characterised in rigour fibres but is also seen in active fibre, but not relaxed fibre. If this pressure-induced ‘expansion’ is equivalent to the length step induced by releasing one end of a fibre, then a quick tension recovery (equivalent to T1–T2 transition in length-step experiments) is expected. (vii).In skinned fibre experiments, a small rapid decrease in tension was observed following pressure release, but no quick recovery of tension was observed. Only the much slower event, which was coupled to Pi release, was seen.(viii).The experiments, where a pressure release is imposed on an intact frog muscle fibre during the tension plateau of a tetanic contraction, attempted to address some of these issues. At 4 °C, a 10 MPa increase in pressure caused ~10% depression of steady tetanic tension that was fully reversible on return to atmospheric pressure (Vawda et al. [7]). This observation is similar to that observed in maximally Ca-activated skinned rabbit muscle fibres at ~12 °C (Fortune et al. [9,22]). With the less compliant intact fibres and a better time resolution, we do now observe a quick tension recovery following the rapid release of pressure that is of the time scale expected for a traditional length step (see Figure 19).(ix).Different rapid perturbations have been used to examine the underlying mechanism of force generation in active muscle. The notion that the force generation step is prior to Pi release (a transition between two AM.ADP.P_i_ states) in the crossbridge cycle is proposed in several studies using different perturbation techniques, such as hydrostatic pressure release (Geeves and Ranatunga, [2]), Pi-jump, (Dantzig et al., 1992 [23]), sinusoidal length oscillation (Kawai and Zhao, 1993 [24]), T-jump experiments (Ranatunga, 1998 [25]) and others. Table 1 provides a brief summary of the experimental data obtained using different perturbations that examined the effects of phosphate on force generation. The data basically show that different perturbations lead to very similar results.

Table 1 provides a brief summary of the experimental data obtained using different perturbations that examined the effects of phosphate on force generation. The data basically show that different perturbations lead to similar results (although the maximum rate is different).

### 4.2. Some Interesting Issues Addressed/Raised by High-Pressure Experiments

#### 4.2.1. Mechanism of Force Generation in Active Muscle

Geeves and Holmes (1999, [28]) suggested that “a change in crossbridge structure pulling on the lever arm may be a possible mechanism of active force development in muscle”. However, such a process is unlikely to absorb heat (not endothermic); also, whether such a process—without change in attachment to thick filament—can generate much force remained unclear (see Sugi, 2017, [29]). Changes in attachments of crossbridge states (non-stereospecific to stereospecific, hydrophilic to hydrophobic, etc.) have been suggested by others (Zhao and Kawai, 1994 [26]: Kodama, 1985 [30]: Ferenczi et al., 2005 [31]) to account for heat absorbing (endothermic) force. Looking back on other ideas, Harrington (1973, [32]); Davis and Harrington (1993, [33]), and Davis and Rodgers (1995, [34]) have discussed several different mechanisms to account for endothermic force in muscle. Davis and Epstein (2007, [35]) proposed that an unfolding within the crossbridge secondary/tertiary structure might cause force generation. Furthermore, they proposed an interesting idea that the forward rate of the force generation step is increased, but the reverse rate is decreased with an increase in temperature. In a recent review, Sugi (2017, [29]) has re-examined the possible importance of a change in crossbridge attachment (subfragment-2) during force generation. It seems **that specific experimental details of an endothermic structural mechanism (with a volume increase) for crossbridge force generation in the muscle that can account also for its coupling to the actomyosin cycle are still lacking (Mobley et al., 2017** [36]). 

#### 4.2.2. Excitation–Contraction Coupling 

As seen in Figure 12b, pressure release during enhanced twitch contraction does not lead to a decrease in the increased force, except when pressure is released very early and close to the stimulus (Figure 12a). This shows that the effect of increased pressure acts on early processes (e.g., Ca release from the sarcoplasmic reticulum) of twitch contraction; this would account for an increased rate of tension rise, increased tension, and slowed relaxation. These effects are basically similar to the tension-potentiating effects produced by caffeine (see Vawda et al., 1995 [7]).

#### 4.2.3. Cause of Muscle Fatigue

Experiments on single intact muscle fibres could be used to examine the process of muscle fatigue and Figure 13 shows experimental traces from one fibre, which also illustrate the effects of increased pressure on fatigue. Figure 13a shows that high pressure (~5 MPa) produces just a small depression (and slowed relaxation) in normal unfatigued fibre. When tetanic contractions are elicited at short intervals to initiate fatigue, muscle tension drops due to fatigue; interestingly, increased pressure leads to tension potentiation in this fatigued muscle, as seen in Figure 13c. It is likely that high pressure enhances Ca release and excitation–contraction coupling.

### 4.3. Some Final Thoughts (in Relation to Wider Knowledge)

**(A)** Based on their length step experiments on muscle, Huxley and Simmons [21] and later Ford et al. (1977 [35], 1986 [37]), interpreted that the quick recovery of tension after a length release represents the crossbridge power stroke and muscle force generation. This quick tension recovery could be separated into two components (see Davis and Harrington, 1993, [33]), fast phase 2a and slower phase 2b; phase 2b was similar to that after a T-jump. By extrapolation, the rate of the phase 2b recovery corresponding to the isometric point in a length step versus the rate of tension recovery plot in experiments on rabbit psoas fibres at ~10 °C is ~40–60 s^−1^ ([27,38]), which is similar to the speed of force rise after a T-jump. Additionally, tension recovery after the length step is faster with P_i_ (Ranatunga et al., 2002, [27]; Nocella et al., 2017, [39]) and slower with MgADP, (Coupland et al., 2005, [38]), as in T-jump tension rise. Hence, tension recovery after a length step has a component homologous to the force rise after a T-jump; this was first proposed and discussed by Davis and Harrington (1993, [33]); see also Davis (1998, [40]). Interestingly, experiments of Gilbert and Ford (1988, [41]) on frog muscle showed that quick tension recovery from length release is associated with heat absorption (i.e., is endothermic). In addition, after a shortening step, the rate of recovery of phase 2 tension is temperature sensitive (Q_10_ of 2–3, Piazzesi et al., 2003, [42]). There is greater temperature sensitivity for phase 2b than there is for the phase 2a component of recovery (Davis and Harrington, 1993 [33]: Davis and Epstein, 2007, [43]). Thus, like the force rise due to a T-jump, phase 2b tension recovery after a shortening step may be an endothermic process. The (fast) phase 2a kinetics after the length release is probably due to some non-crossbridge viscoelasticity within muscle fibres, as suggested in several studies (Davis and Harrington, 1993 [33]; Davis, 1998 [40]: Sugi and Kobashil, 1983 [44]). Interestingly, the initial fast tension recovery (as after L release), was seen after a stretch in modelled tension responses to length perturbations (Offer and Ranatunga, 2013, [45]).

Developing a model to simulate the steady-state force–velocity data, L-step force transients, crossbridge stiffness, and energetics in active muscle, we found that having two force generation steps (of similar magnitude) in a simple, unbranched, linear crossbridge cycle are adequate (see Offer and Ranatunga, 2013, [45]). Our conclusion contrasted with other complex schemes proposed from X-ray studies of shortening muscle by Huxley et al. (2006, [46]); they suggested a working stroke of a crossbridge has several molecular steps. When the model developed with two force-generating steps [45] was used to simulate the effects of temperature, Offer and Ranatunga, (2015, [47]), it was found that the endothermic ATP hydrolysis and crossbridge attachment steps (as found in biochemical studies) contributed to the tension increase in isometric and shortening muscle; also, the first force-generation step was endothermic [1]. 

As shown in the data presented here using hydrostatic pressure effects, a force rise on the release of hydrostatic pressure (indicating a volume increase) has been reported in experiments on skinned rabbit fibres (Fortune et al., 1991, [22]) and on intact frog fibres (Vawda et al. 1999, [20]). Interestingly, and according to our thinking, an endothermic (heat-absorbing) force development in attached crossbridges could account for such an effect.

**(B)** In consideration of skeletal muscles, their variability in structure and performances in our body, in different species and at different ages, there are many other experimental observations/issues that a future student ought to keep in mind. A few of these issues are listed briefly below.

(i).For instance, as shown experimentally (see Ranatunga and Thomas [48] and refs therein) muscle structure, biochemistry, and contraction characteristics from the same animal can be different and show different types, fast and slow, or types 1, 2, etc.(ii).There have been views on muscle structure, sarcomere changes, etc. that raised issues which are not fully understood and agreed on at present [49,50,51].

## Figures and Tables

**Figure 1 ijms-24-05031-f001:**
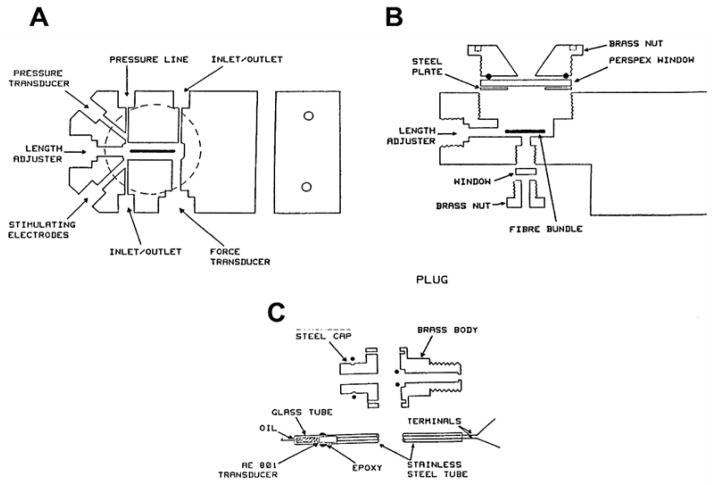
Schematic diagrams of the pressure chamber. (**A**) Top view of a cross-section showing ports. (**B**) A vertical cross-section showing the two Perspex windows. Muscle fibre bundle is shown by a dark line. (**C**) The force transducer plug: an AE 801 transducer element was glued to a stainless steel tube that could be pressure-sealed; the beam of the AE 801 element was enclosed in a glass tube and filled with paraffin for electrical insulation, when transducer was within chamber. (Filled circles show positions of some of the rubber O-rings for pressure sealing, adapted from Ranatunga et al. [8]).

**Figure 2 ijms-24-05031-f002:**
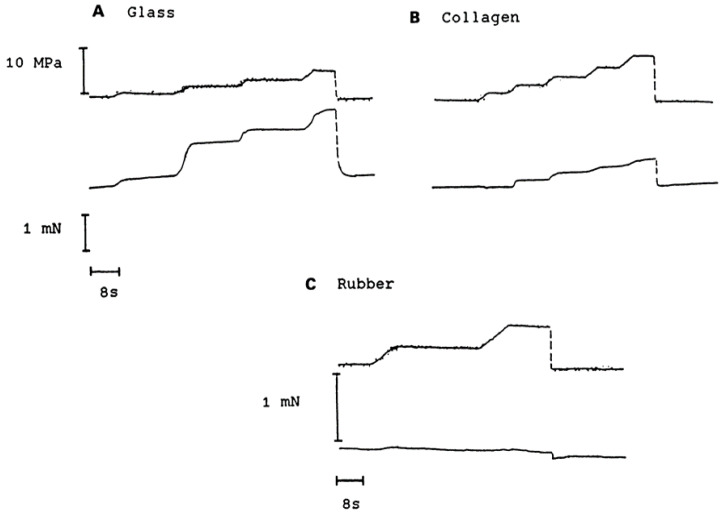
Experimental tension records from glass (**A**), collagen (**B**), and rubber fibres (**C**). In each panel, pressure trace is shown on top of the tension trace and position of the horizontal time scale bar denotes zero tension. Note that isometric tension of glass and collagen increases reversibly with increased pressure; tension of rubber is insensitive to pressure (from Ranatunga et al. [8]).

**Figure 3 ijms-24-05031-f003:**
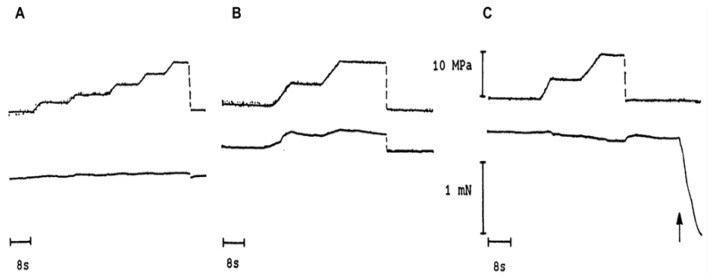
Tension records (lower trace in each) from a muscle fibre bundle (5 fibres, cross-section of 120 um and length of 4 mm) exposed to different pressures (upper trace in each). In each frame, the position of the horizontal time bar represents zero tension. (**A**) in relaxed state, (**B**) in rigour state, and (**C**) in maximally Ca-activated state—bundle was relaxed at the upward arrow, by exposing to relaxing solution (-no Ca). (Adapted from Ranatunga et al., 1991, [8]).

**Figure 4 ijms-24-05031-f004:**
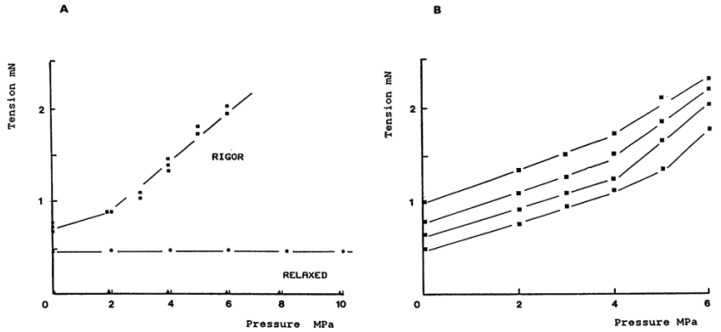
Plots of tension versus pressure from one fibre bundle (diameter 280 µm and length 5 mm). (**A**) Fibre bundle in relaxed and rigour. Relaxed data are averages from three separate pressure perturbations and the range was within the diameter of the symbol: those for rigour are individual measurements. (**B**) Data from the same preparation at different initial rigour tensions; note that pressure/tension plots are approximately parallel.

**Figure 5 ijms-24-05031-f005:**
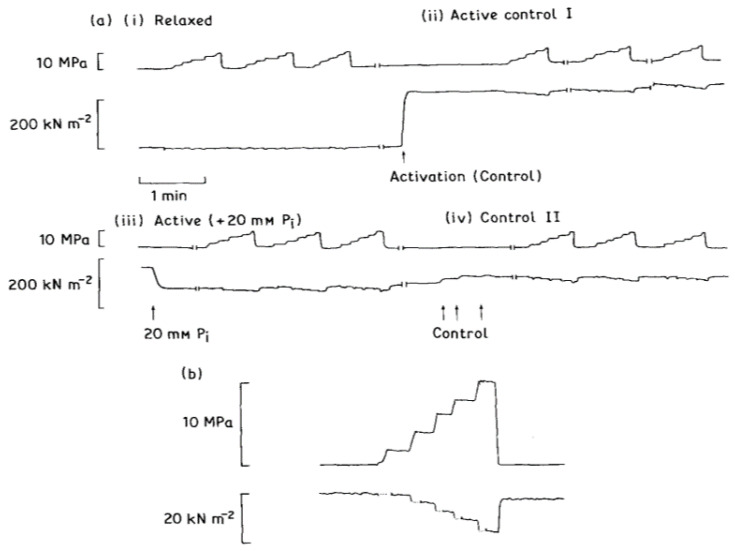
(**a**) Strip chart record showing the experimental protocol: the upper trace shows pressure and the lower trace, tension. The pressure sensitivity of tension was first examined when the fibre was relaxed (**i**). The fibre was then maximally Ca-activated in control solution (no Pi) and again the effect of pressure recorded (**ii**). Taking this to be zero time, the response to pressure was then recorded in test solution (+ 20 mM Pi) after 14 min (**iii**) and again in control solution (**iv**) after 22 min. Arrows indicate the points at which the solution was changed. Note that the sections illustrated are from a continuous record: the steady tension was depressed by 40% on exposure to Pi and then recovered to 80% of the original tension after Pi removal. Under each condition (**i**–**iv**) the steady tension at high pressure was normalized to the mean tension recorded at atmospheric pressure immediately before and after a series of pressurizations. (**b**) A computer stored records of tension and pressure during an active contraction; the actual measurements were made from them (taken from Fortune et al. [9]).

**Figure 6 ijms-24-05031-f006:**
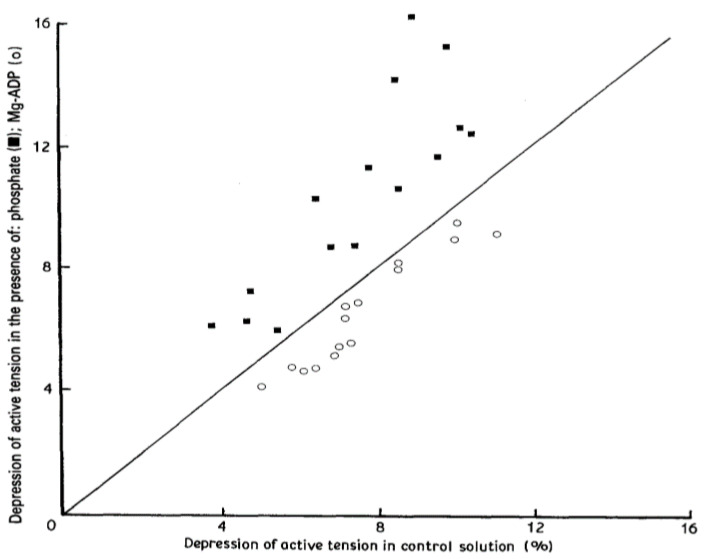
Comparison of the pressure-induced tension depression in the presence of 20 mM phosphate and in the presence of 11 mM Mg^2+^-ADP. The percentage tension depression induced by 10 MPa pressure in the control solution (i.e., in the absence of phosphate or Mg^2+^-ADP) is plotted on the abscissa against the corresponding value in either phosphate (■) or Mg^2+^-ADP (○) (ordinate). Note that all the points in the presence of phosphate (■) are above the line of identity (full black line) whereas those in the presence of Mg^2+^-ADP (○) below it. (Adapted from Ref [9]).

**Figure 7 ijms-24-05031-f007:**
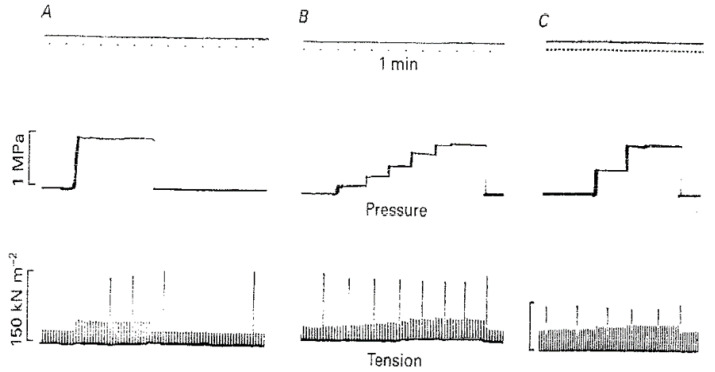
Chart recordings of tension (lower trace) and pressure (upper trace): (**A**,**B**) are from one fibre preparation ((**C**) from another), twitch contractions at 1/10 s and tetani infrequently. Amplitude of twitch is higher whereas that of tetani lower at high pressure (adapted from [18]).

**Figure 8 ijms-24-05031-f008:**
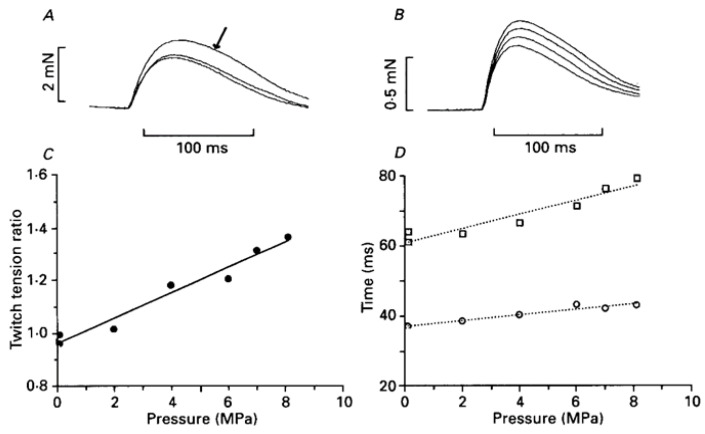
(**A**) Three twitch contractions recorded at 1-minute intervals from an intact fibre bundle: pressure in the chamber was raised to 5 MPa for about 1 min when the second contraction was elicited (second is indicated by arrow). (**B**) Twitches from another (width 140 µm) were recorded when pressure was increased in 2 MPa steps from atmospheric to 6 MPa. (**C**) Pressure dependence of tension and (**D**) time to peak (TP) and time to half relaxation (HR) of twitch contraction; lines are linear regressions (*p* < 0.01) (adapted from Ranatunga and Geeves, 1991 [18]).

**Figure 9 ijms-24-05031-f009:**
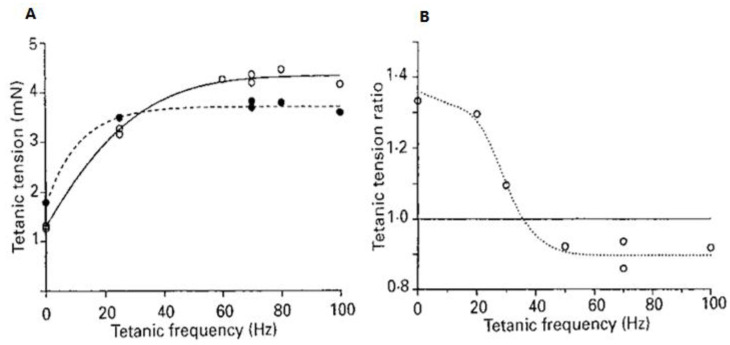
*(***A** left), Stimulation frequency versus tension at atmospheric pressure (○) and at a higher pressure (5 MPa, ●) from one fibre bundle (width 300 µm). (**B** right), From another fibre bundle, tetanic tension recorded at 10 MPa plotted (as a ratio of that at atmospheric pressure) against frequency; data at frequency 0 Hz represent twitch (adapted from Ranatunga and Geeves, [18]).

**Figure 10 ijms-24-05031-f010:**
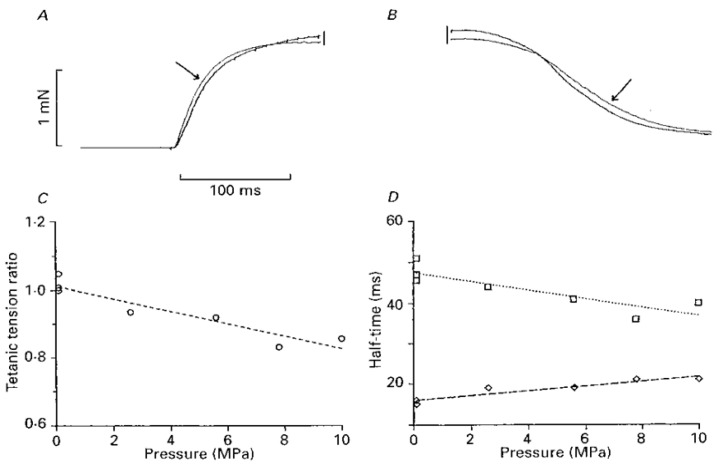
*(***A**,**B**) Tetanic contraction at 6.5 MPa (arrowed) superimposed on that at atm pressure: A and B show the rising and falling phases. (**C**,**D**) Pressure dependence of tension (**C**) and half-time to tension rise (□) and half-time to tension relaxation (◊) (adapted from [18]).

**Figure 11 ijms-24-05031-f011:**
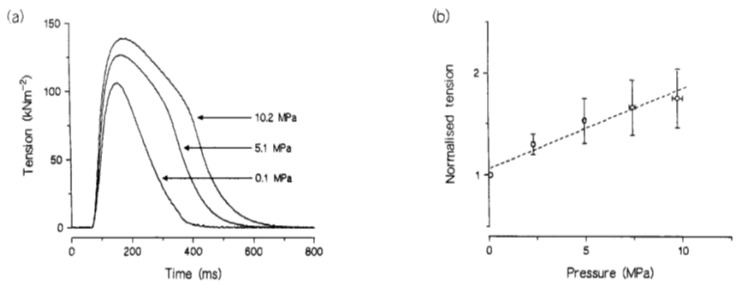
Effects of high pressure on isometric twitch contractions of a single fibre at 4 °C. (**a**) Sample records at three pressures, (**b**) pooled data from four fibres (from Vawda, Geeves, and Ranatunga, [7]).

**Figure 12 ijms-24-05031-f012:**
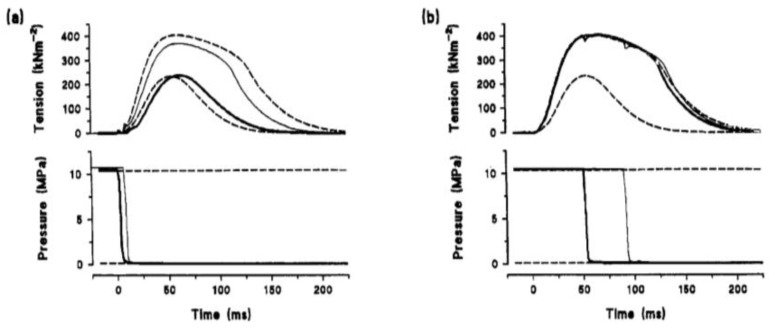
Rapid pressure release from 10 MPa to atmospheric. (**a**) Pressure release during the early part of the twitch and (**b**) during later times. Stimulus is at time = 0: contractions at atmospheric and high (10 MPa) steady pressures are shown by interrupted lines. Note that when pressure is released at 1.5 ms after stimulus, twitch contraction remained similar to atmospheric (unaffected by high pressure).

**Figure 13 ijms-24-05031-f013:**
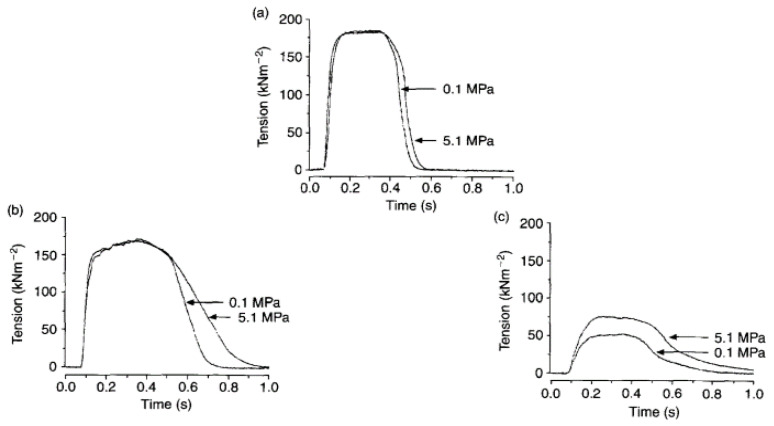
Pressure effects on tetanic contractions at different stages of fatigue (Temp 10 °C). (**a**). before fatigue and (**b**) after 5 min into fatigue: high pressure caused a slight decline in tetanic tension. (**c**) After 10 min into fatigue, tension declined to 30%, and pressure potentiated the tension (Vawda et al., 1996, [12]).

**Figure 14 ijms-24-05031-f014:**
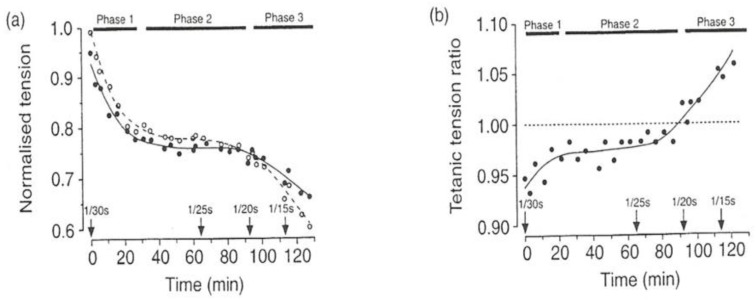
Data from one fibre showing fatigue-induced tension decline: horizontal bars (on top) indicate the phases of fatigue, as in Lannergren and Westerblad, (1991, [19]). (**a**) Open symbols are at atmospheric pressure and solid symbols are at high pressure (10 MPa). (**b**) Same data plotted as a high/atmospheric pressure tension ratio: note that tetanic tension is depressed by high pressure in phases 1 and 2 but potentiated at phase 3.

**Figure 15 ijms-24-05031-f015:**
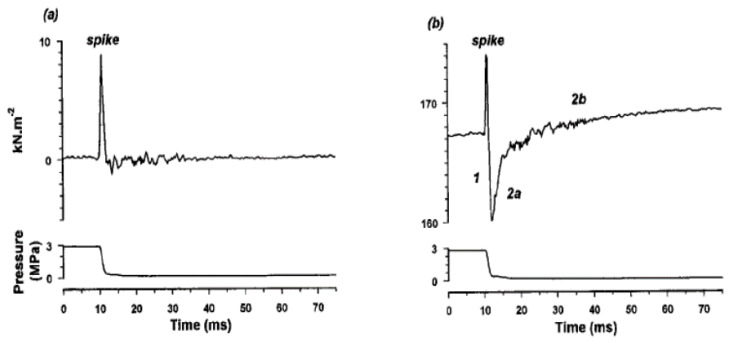
Tension transients induced by rapid pressure release, from a single muscle fibre at ~4 °C (the steady hydrostatic pressure was increased to ~3 MPa and rapidly released (<1 ms) to atmospheric level). (**a**) The resting tension is largely unaffected by the perturbation; the spike on the tension trace is probably a transducer response to rapid decompression. (**b**) Response of the active fibre: pressure release made on the plateau of the tetanus. In addition to the spike (as in a) is a tension drop (Phase 1), followed by a rapid recovery and rise (Phase 2a and 2b) to a level higher than that prior to release (from Vawda et al., 1999, [20]).

**Figure 16 ijms-24-05031-f016:**
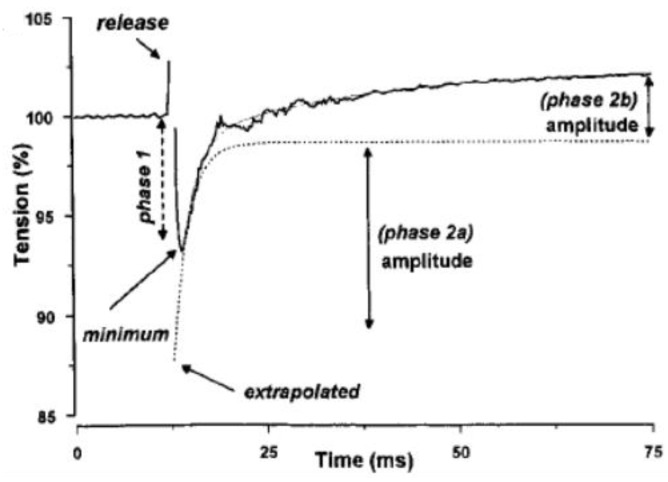
The tension transient to a pressure release in an intact fibre. A double exponential curve can be fitted to the tension recovery. The calculated rates for Phases 2a and 2b were ~500 and 50/s, (adapted from Vawda, Geeves, and Ranatunga, 1999, [20]).

**Figure 17 ijms-24-05031-f017:**
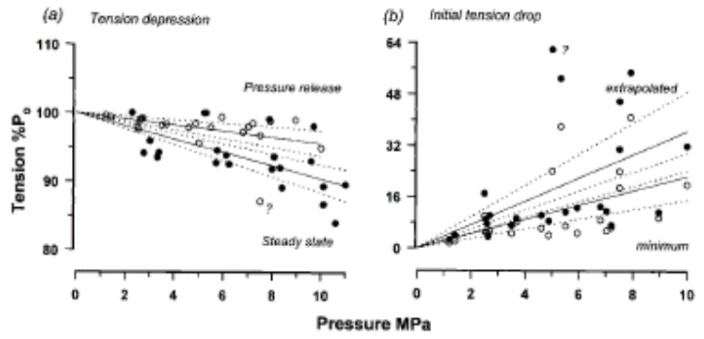
Pressure sensitivity of tension components: abscissa is the hydrostatic pressure to which a fibre was exposed prior to the release. (**a**) Filled circles: tetanic tensions measured from contractions at different pressures (the solid line is the fitted linear regression and the dotted lines—95% confidence envelope, *p* < 0.01, n = 21). The tension depression is 0.96% per MPa. Open circles: the steady tetanic tension level (high-pressure tension) prior to release is plotted as a percentage of the steady tension level (atmospheric) reached after the release. The linear regression fitted to the data, constrained to pass through 100% tension at zero MPa, is 0.44%/MPa. (**b**) The initial tension drop (P_1_), plotted as a percentage of the recovered (atmospheric) tetanic tension. Open circles are the values measured from tension records and filled circles show the tension drop estimated by curve fitting. The tension drop is significantly correlated (*p* < 0.05) and is 2.25% per MPa for measured and 3.35% per MPa for the extrapolated (the extreme high value, labelled with a question mark was excluded in both (**a**,**b**)).

**Figure 18 ijms-24-05031-f018:**
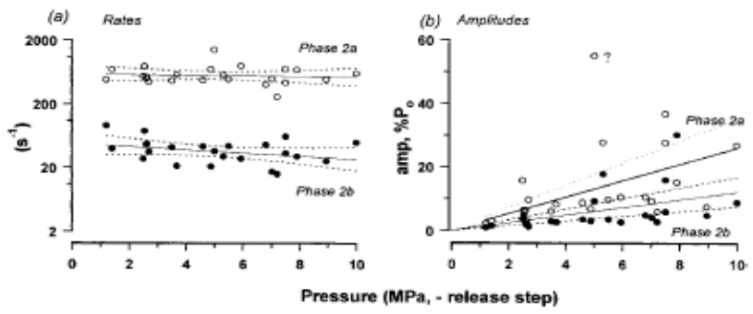
(**a**) The rates of the two exponential components (Phases 2a and 2b) are not correlated with pressure-jump amplitude (*p* > 0.1); the mean rate was ~600/s for Phase 2a and ~40/s for Phase 2b **20.** (**b**) The amplitudes of the two components (as % tetanic tension), increase with amplitude of release (from Vawda, Geeves, and Ranatunga, 1999 [20]).

**Figure 19 ijms-24-05031-f019:**
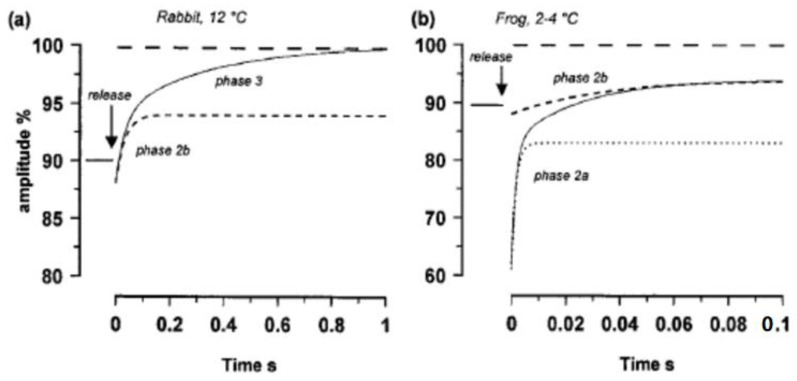
(**a**) The tension transient of a maximally Ca-activated rabbit muscle fibre at 12 °C (as in Fortune et al. 1994). A fibre was maximally Ca-activated at atmospheric pressure, the pressure was then increased to 10 MPa after maximal active tension and tension was depressed to ~90%. The recovery after rapid release occurred in two phases (Phase 2 at ~30/s and Phase 3 at ~3/s). The initial tension drop (Phase 1) was small (<5%). Phase 2 (dashed curve) was sensitive to inorganic phosphate and was identified as crossbridge force generation (see Fortune et al., 1991, [22]) and Phase 3 was identified as the rate-limiting step in crossbridge cycle (adapted from Vawda et al., 1999, [20]). (**b**) The tension is transient from frog muscle fibre at 2–4 °C (Vawda et al., 1999, [20]) (the pressure was increased to 10 MPa when the fibre was relaxed, the fibre was activated by electrical, tetanic, stimulation and released during the tension plateau). In comparison with the tension of an atmospheric contraction, the tension at 10 MPa was lower, ~90%, (see Figure 3a) and the tension recovery after pressure release was incomplete within the duration of the tetanic tension plateau (<200 ms). The transient has two recovery phases (Phase 2a, ~500/s and Phase 2b ~40/s) and the initial tension drop upon pressure release is 30%. The Phase 2a does not contribute to tension rise above the pre-release level (dotted curve). The short dashed curve is Phase 2b in the rabbit fibre.

**Table 1 ijms-24-05031-t001:** Effects of phosphate (Pi) on force generation in skeletal muscle.

Perturbation	Min. Rate	Max. Rate (s^−1^)	K_d_ (mM)	Q_10_	Source
*Hydrostatic pressure*	17	52	4	~4	Fortune et al., 1991 [22]
*Sinusoidal length*	~20	~80	4	4	Zhao and Kawai, 1994 [26]
*Phosphate release*	21	123	12	3.7	Dantzig et al., 1992 [23]
*Temperature jump*	50	185	9	~4	Ranatunga, 1996 [19,25]

The above data are from experiments at ~10–12 °C. The observed rate increased with [Pi] and the data give the minimum and maximum rate constants determined by hyperbolic curve fit to the observed rates (phase 2b) at various [Pi]. [Pi] for half-maximal change is given as K_d_. Q_10_—temperature coefficient (adapted from Ranatunga et al., 2002 [27]).

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
