# Peer review of "Effects of Hydrostatic-Pressure on Muscle Contraction: A Look Back on Some Experimental Findings"

_ijms, 2023, doi:10.3390/ijms24055031_

Round 1

Reviewer 1 Report

The article ”Effects of Hydrostatic-Pressure on Muscle Contraction: A Look-back on Some Experimental Findings” explains possible underlying mechanisms of tension potentiation and causes of muscle fatigue in correlation with hydrostatic pressure. As improvement suggestions are:

1. The level of self-citation seems to be high, with 9 out of 51 citations. Searching for the same title words on google, I obtained about 36,700,000 results. 

2. Figures can be improved in order to be easily understood.

3. Will be interesting to be discussed the clinical/therapeutical relevance of hydrostatic pressure in rehabilitation applications.

Overall, the article is of high scientific quality.

Author Response

Thank you for your valuable observations1

I have now added some other references that may be of interest to the readers! This reduces self citation!

Also, some of them provide  interesting facts regarding pressure effects.

Reviewer 2 Report

This is an interesting and generally well-written review (or mix btw review and original paper) by Ranatunga and Geeves about how increases in pressure from atmospheric (0.1 MPa) to high values (10 MPa) affect muscle fibers. The authors largely refer to their own original work which is reasonable considering that a search on PubMed shows that they have co-authored a dominant fraction of the existing papers in the field. The review summarizes previous studies quite comprehensively. It further discusses mechanistic insights related to the force-generating structural changes in actomyosin, excitation-contraction coupling and mechanisms of fatigue.

Whereas I think that this is a nice study that deserves to be published I have some suggestions for the authors as follows:

1. I strongly suggest that the authors add a paragraph in the Introduction (or a separate Theory section) to make the overall subject a bit easier to grab for the average muscle researcher. Such a paragraph should define some concepts such as rubber-like elasticity, normal elasticity etc. and how materials with these properties are known to respond to increased pressure and why. Also, in such a paragraph it would be of value with some general information about what type of mechanistic information about muscle and protein function that one can expect to gain from studies at varied pressure. In this connection it would be of value with something about the most central thermodynamic relationships between free energies, pressures and volumes.

2. References and extended discussion

-Lines 28-29: “by Ranatunga [1] and the references therein” I suggest adding one or more general and recent reviews because that of Ranatunga primarily focuses on temperature effects.

-Lines 431-436: It should be mentioned that there are also potentially appreciable challenges with the idea of force-generation before Pi-release (e.g. with references to Moretto et al. Nature Commun 2022 and possibly other studies).

-Lines 505-507: There are arguments against the views expressed here, based on time-resolved X-ray muscle fiber diffraction and fluorescence polarization data of Lombardi, Irving and co-workers from the late 1990s and early 2000s. In these studies they find quite convincing evidence for lever arm swing during phase 2a in active muscle whereas they do not see similar events in rigor. Suitable citation(s) of these authors should be given to expose this alternative view.  

-Somewhere in the Discussion: Relate to some physiological findings. I understand that the pressure perturbation studies described here do not primarily aim to elucidate normal physiological response but rather to provide insights into basic mechanisms. Nevertheless, it may be of interest to consider the results in relation to some physiological data, e.g. Rossignol et al Comparative Biochemistry and Physiology A, 2006 and Sleboda and Roberts PNAS, 2019.

3. Clarity of presentation and language

(With regard to the language, I am not a native English speaker so my comments may not be fully appropriate. In that case, please forgive me. However, I nevertheless wish to point out some lines that appear odd or confusing to me.)

-Line 41. Maybe change "not fully” to "incompletely" at least in some of the occurrences

-Line 61. “isolated actin and myosin (sub-fragment),”  possibly skip parenthesis and write myosin motor fragments or something similar.

-Line 68. “in such studies” -> “in such pressure perturbation studies” for clarity

-Line 76. “stainless block” ->“stainless steel block”?

 -Line 83. “was for temperature. The”-> “was for temperature control/monitoring. The”?

-Line 117. “the pressure effects in relation to some are listed” It is not entirely clear to me what is meant.

-Lines 133-134. “and not just in the series-elasticity in them” ->and not just in the series-elasticity” Would it be possible to simplify as suggested?

-Line 169. “kinetics of actomyosin interaction (10)”->” kinetics of the actomyosin interaction (10)”?

-Legend Fig 5, Lines 192-193. “(b), A computer stored record of tension and pressure from which such measurements were made (adapted from Fortune et al., [9]).” I do not understand how this text relates to panel b?

-Fig. 6. I presume that the spread along the horizontal axis is the normal variation in tension depression upon increase from atmospheric to 10 MPa pressure? Is this variability just a random effect or can the authors identify any underlying causes?

-Legend of Fig. 7. This legend should be made a bit more informative.

i. Panels A and B are mentioned but what about panel C?

ii. Does 1 min refer to the distance between the dots or what does it stand for?

iii. The dots appear more frequently in C but there does not seem to be any corresponding change in the distances between twitches in the bottom trace, or?

-Line 224. “high pressure, is accompanied” -> “high pressure, accompanied”

-Legend Fig. 8, Line 229. “(for about 1min) when the second contraction elicited (indicated by arrow).” I am not quite sure what the authors mean here.

-Lines 235-236. “are not summated, remain as twitches, the tension is potentiated,” It is not clear to me what the authors intend to say here.

-Line 259. “Exact nature” -> “The exact nature”?

-Line 262. “records illustrated in Figure”  What figure?

-Lines 267 and 269. Change order of panels 12a and 12b to mention them in logical order.

- Fig. 14. What do 1/30s etc. stand for?

-Line 330. “Data in Figure 17a from show that the” Something seems to be missing.

-Figs. 17 and 18. Poor resolution

-Line 445. “perturbations lead to very similar results.” It may nevertheless be worth to comment on the almost 4-fold differences in maximum rate

-Line 451. “Such a process is unlikely to be “heat absorbing (endothermic)”  Please elaborate.

-Line 453. “force also remains unclear” -> “force remains unclear”

-Lines 464-466. “It seems that specific experimental….acto-myosin cycle are still lacking.”  

Please elaborate a bit more also on this! For instance, is it not so that it is not entirely self-evident how a volume increase could be brought about in reality (cf. Akasaka and Maeno Biology 2022)

Author Response

Thanks for your useful comments.

I have made changes as suggested, but not all, given that some changes were suggested by others.

Hope this is acceptable.